



# Hyper-resolution ensemble-based snow reanalysis in mountain regions using clustering

Joel Fiddes[1,2], Kristoffer Aalstad[1], and Sebastian Westermann[1]

[1]University of Oslo
[2]WSL Institute for Snow and Avalanche Research SLF

*Correspondence to:* Joel Fiddes (joel.fiddes@slf.ch)

**Abstract.** Spatial variability in high-relief landscapes is immense, and grid-based models cannot be run at spatial resolutions to explicitly represent important physical processes. This hampers the assessment of the current and future evolution of important issues such as water availability or mass movement hazards. Here, we present a new processing chain that couples an efficient subgrid method with a downscaling tool and data assimilation method with the purpose to improve numerical simulation of surface processes at multiple spatial and temporal scales in ungauged basins. The novelty of the approach is that while we add 1-2 orders of magnitude of computational cost by ensemble simulations, we save 4-5 orders of magnitude over explicitly simulating a high resolution grid. This approach makes data assimilation at large spatio-temporal scales feasible. In addition, this approach utilises only freely available global datasets and is therefore able to run globally. We demonstrate marked improvements in estimating snow height and snow water equivalent at various experimental scales using this approach. We propose this as a suitable method for a wide variety of operational and research applications where surfacecheck models need to be run at large scales with sparse to non-existent ground observations and with the flexibility to assimilate diverse variables retrieved by EO missions.

## 1 Introduction

Accurate simulation of energy and water cycles in high mountain environments is critical for a wide range of operational and research applications related to water resources and natural hazards, particularly in the current era of dramatic changes in mountain regions worldwide (Mankin et al., 2015). However, basic surface variables in many remote mountain areas remain poorly quantified despite large increases in the capacity of in-situ observations, remote sensing platforms and atmospheric model products. Spatial resolutions of 100 m are commonly recommended for modelling of land surface variables such as snow cover or surface temperature in complex terrain (Bierkens et al., 2015; Wood et al., 2011; Baldo and Margulis, 2018) and has come to be known as hyper-resolution (Wood et al., 2011). This is due to the fact that energy and mass fluxes exhibit strong lateral variation due to the effects of topography (Gruber S. and Haeberli W., 2007), and surface/subsurface properties such as vegetation cover (Shur and Jorgenson, 2007), ground material (Gubler et al., 2012) or snow distribution (Zhang, 2005; Liston, 2004) further compound these effects.





Most continental to global modelling studies operate on regular grids which has placed limitations on model resolutions despite advances in computing power. However, previous efforts using Hydrological Response Units, HRUs (Beven and Kirkby, 1979; Durand et al., 1993; Fiddes and Gruber, 2012), triangular irregular networks (Mascaro et al., 2015; Tucker et al., 2001), or multi-resolution approaches (Baldo and Margulis, 2018) suggest that regular grids are not only expensive but sub-optimal as often only subsets of watersheds require detailed model descriptions in order to characterise the system adequately. In addition, deterministic modelling schemes have limitations even at hyper-resolution due to errors in forcing data, particularly with fields such as precipitation which suffer from both measurement and modelling biases. The numerical weather prediction community has been addressing this problem for several decades using various data assimilation (DA) approaches. DA methods, often with Bayes' rule as a starting point, attempt to ingest uncertain observations into uncertain model simulations (Lahoz and Schneider, 2014; Carrassi et al., 2018). It is a class of methods that are implicitly Bayesian in that uncertainty in both simulation and observation are accounted for. These methods are diverse in design and application and the reader is directed to Liu et al. (2012) for a review relevant to the land surface community or Carrassi et al. (2018) for a timely overview. Only relatively recently has data assimilation started to be utilised in land surface modelling schemes (Liu et al., 2012), but it has already shown much promise in the current era of plentiful remote sensing data. Recently, ensemble-based DA has been successfully applied to the problem of improving snowpack estimates at various spatial scales (Margulis et al., 2015; Aalstad et al., 2018; Magnusson et al., 2017; Griessinger et al., 2016), this is particularly pertinent as it is widely recognised that estimating the spatial distribution of snow water equivalent (SWE) in mountain regions is currently one of the most important unsolved problems in snow hydrology (Dozier et al., 2016) and in understanding spatial distribution of other processes dependent on the snowpack mass balance, such as the surface energy balance.

Ensemble-based data assimilation revolves around the use of an ensemble (i.e. a collection) of model trajectories. Each trajectory is referred to as an ensemble-member or particle, for economy we will use the latter. An ensemble allows for the quantification of uncertainty through the prior (before assimilation) and posterior (after assimilation) distribution of particles. The use of an ensemble increases the computational burden, often adding orders of magnitude to computation times. Given that computation time is practically limited, in ensemble-simulation there is always a trade-off between a model's spatio-temporal resolution and the number of particles. Both are desirable, given that higher spatio-temporal resolution (is expected to) increases model realism whereas a higher number of particles allows for improved uncertainty estimation. This is why the dual quest for efficiency in models and DA is important. We argue that sometimes some of the resources that are spent on explicit high resolution spatial modelling could be better spent on the ensemble. When discussing computational expense it's worth noting that the intended application is important to consider. Given a large HPC infrastructure and enough time, today, we have the ability to use brute force deterministic numerical simulations to solve many resource intensive problems. However, the question is (a) what better purposes could that computation time be used for (e.g. uncertainty quantification) and (b) are we producing a final product (where one off large simulations are tolerable) or as is more commonly the case, at least in research (but also operational centres), are we part way through a development cycle where we expect to make many iterations in order to gain knowledge of the system. In this second case there is a strong motivation for methods that allow quick development cycles and knowledge gain. The previously published TopoSUB and TopoSCALE models (Fiddes and Gruber,





2012, 2014) are hyper-efficient approaches which may provide a solution for this problem, particularly in data sparse regions. TopoSUB is a subgrid method that permits order of magnitude efficiency gains in applying numerical models over large areas. It achieves this by using a multivariate clustering of input predictors (normally topographical parameters) to reduce the number

of simulations required to accurately represent surface heterogeneity by orders of magnitude. TopoSCALE provides point scale meteorological forcing at any given point on the earths surface by downscaling gridded reanalysis (or other atmospheric model data) using pressure levels to account for gradients with elevation and topographic correction for surface energy balance terms. The computational resources saved by not simulating domains explicitly in 2D can then be redirected to ensemble simulation for the purpose of data assimilation or uncertainty analysis in general. This approach has successfully been used to generate a

regional scale permafrost map at 30 m resolution (Fiddes et al., 2015).

In this paper we present a new processing chain that couples an efficient subgrid method (TopoSUB), a downscaling tool (TopoSCALE) and data assimilation method with the purpose to improve numerical simulation of ground surface processes at multiple spatial and temporal scales in ungauged basins. The novelty of the approach is that while we add 2 orders of magnitude of computational cost by ensemble simulations, we save 4-5 orders of magnitude over explicitly simulating a high resolution

grid. This approach makes data assimilation at large spatio-temporal scales feasible. In addition, this approach utilises only freely available global datasets and is therefore able to run globally.

Applications of this approach are numerous and diverse as it addresses 3 common bottlenecks: (a) availability of an appropriately downscaled forcing (b) ability to apply complex models at high resolution over large areas and (c) addressing uncertainty in the model chain. Applications could for example include large scale assessments of mass movements, glacier mass balance,

or snowpack water availability. By translating GCM/RCM results to local slope scale impacts with appropriate surface models, climate change impacts can be estimated at appropriate scales.

## 2   Methods

The modelling pipeline used in this study employs two previously described methods (1) TopoSUB (Fiddes and Gruber, 2012) and (2) TopoSCALE (Fiddes and Gruber, 2014). These tools are briefly described here for clarity, however the reader is directed

to the original publications for full details. An overview of the full tool chain is given in Figure 1.

### 2.1   Surface model

The surface model used in this study, GEOtop, is a physically-based model originally developed for hydrological research (Endrizzi et al., 2014). It couples energy and water budgets, represents the energy exchange with the atmosphere and has a multilayer snow pack. Further information is given by Bertoldi et al. (2006); Rigon et al. (2006); Endrizzi (2007); Dall'Amico

et al. (2011). A description of model uncertainty and sensitivity is given by Gubler et al. (2012). Model parameters and soil stratigraphy are setup as defined in Fiddes et al. (2015).



## 2.2 Downscaling forcing

TopoSCALE is a scheme which generates point-scale model forcing using gridded atmospheric model datasets. It achieves this as follows: (1) interpolate data available on pressure levels: air temperature ($T_a$), relative humidity ($RH$), wind speed ($U$), wind direction ($\varphi_U$) to point of interest in order to provide a dynamic scaling at each timestep, (2) incoming longwave radiation ($L^\downarrow$) is scaled by accounting for downscaled $T_a$, $RH$ and sky emissivity; (3) we apply a topographic correction to both radiation fields ($S^\downarrow/L^\downarrow$); (4) an elevation based lapse-rate is applied to precipitation, $P$. The output is a full set of scaled meteorological fields required to drive a numerical model at hourly timesteps.

## 2.3 Subgrid scheme

TopoSUB is a scheme which samples land surface heterogeneity at high resolution based on a DEM and other surface data (here SRTM-3, 30 m). Input predictors describing dimensions of variability are clustered with a K-means algorithm to reduce computational units in a given simulation domain to a set of clusters. A 1-D surface model is then applied to each cluster using its mean physiographic properties. This approach allows multiple orders of magnitude savings in computational effort over distributed approaches. For example, a simulation domain represented by an ERA5 grid cell (25 km × 25 km) contains approximately $10^6$ SRTM-3 pixels. This domain can be simulated using 100 TopoSUB clusters, which represents a $10^4$ reduction in computational load during simulation.

## 2.4 Data assimilation

We build on previous efforts (e.g. Girotto et al., 2014; Margulis et al., 2015; Aalstad et al., 2018) that focus on the reanalysis of snowpack characteristics (particularly SWE and HS) through ensemble-based assimilation of fractional snow covered area (fSCA) retrievals from optical satellite sensors. We choose to use fSCA retrievals because currently only optical satellite sensors can offer the resolution, coverage, accuracy and breadth of information needed to constrain snowpack simulations in complex terrain (see Dozier et al., 2016). We use fSCA retrieved from the MODIS sensors onboard the Aqua and Terra satellites. These retrievals have a sub-kilometric spatial resolution and a near daily equatorial revisit frequency (in the absence of clouds), so the reanalysis we perform could be applied to any mountain range on Earth. By assimilating fSCA observations we exploit the dynamic information content contained in the depletion of the fractional snow-cover. The idea is that if one grid-cell melts out later than another, there must either have been more snow there to begin with, a slower ablation, or a combination of the two and vice-versa for an earlier melt out (Aalstad et al., 2018). This is the essence of traditional snow reconstruction where the snowpack is built up in reverse from the observed date of disappearance of the snow-cover to the day of peak SWE using modelled snowmelt rates (Martinec and Rango, 1981; Dozier et al., 2016). By using ensemble-based DA we can account for uncertainties in the remotely-sensed fSCA depletion, the meteorological forcing and the snow model that are ignored in traditional reconstruction (Slater et al., 2013) and arrive at an improved reanalysis (Girotto et al., 2014). Snow reanalysis problems are best approached using batch smoother DA algorithms rather than the more commonly used filters since


the snowpack has a long memory (i.e. high temporal autocorrelation) relative to (e.g.) synoptic-scale weather (Margulis et al., 2015; Aalstad et al., 2018). By using a smoother that assimilates all the fSCA retrievals during the ablation season at once to constrain the ensemble of annual snowpack trajectories, we are able to use the observed ablation to inform the accumulation season which would not be possible with a particle filter.

### 2.4.1 Generating the prior ensemble

In line with previous studies (e.g. Raleigh et al., 2015), we assume that the main source of uncertainty in modelling the snowpack is in the meteorological forcing and specifically the main variables that control the mass and energy balance, namely air temperature ($T_a$), precipitation ($P$), incoming shortwave ($S^{\downarrow}$) and longwave ($L^{\downarrow}$) radiation. To generate the prior ensemble we perturb the forcing time series using normally ($T_a$, $S^{\downarrow}$, $L^{\downarrow}$) and log-normally ($P$) distributed multiplicative perturbation parameters that are fixed throughout the annual integration. Following Navari et al. (2016) we generate a correlated ensemble

of perturbation parameters for the different forcing variables. This is to avoid unrealistic perturbations such as a large increase in both precipitation and shortwave radiation. We do this in two steps. First, generate independent perturbation parameters for each of the forcing variables using normal and lognormal random draws. Secondly, we account for the correlation between the different perturbation parameters by performing a Cholesky decomposition of the covariance matrix. All hyper-parameters used in generating the prior ensemble are given in Table 1.

### 2.4.2 Particle batch smoother

When performing DA we are usually interested in approximating the Bayesian posterior: the probability of model trajectories given the observations. The DA method employed in this study is the particle batch smoother (PBS) presented in the context of snow reanalysis in Margulis et al. (2015). The PBS is a basic importance sampling particle filter where no resampling takes place (see Van Leeuwen, 2009). This means that it is equivalent to the generalized likelihood uncertainty estimation

(GLUE) with a formal likelihood function (Beven and Binley, 1992). The apparent advantage of this smoother is that, unlike the ensemble smoother (ES), it makes no assumptions about the linearity of the model or the Gaussianity of the error statistics (Van Leeuwen and Evensen, 1996). This can also be a disadvantage in higher dimensional problems where the method is prone to degeneracy and large sampling error unless a very large number of particles is used (Van Leeuwen and Evensen, 1996; Van Leeuwen, 2009). Nonetheless, for snow reconstruction problems where the dimensionality of the parameter space is

relatively low, the PBS has been shown to outperform the ES even with a moderate number of particles (Margulis et al., 2015; Aalstad et al., 2018). Crucially, using the PBS instead of the ES (or its iterative variants) avoids the need for running more than one ensemble model integration, which would be more costly and difficult to reconcile with the clustering (TopoSUB) framework. Since the PBS is derived elsewhere (Van Leeuwen and Evensen, 1996; Van Leeuwen, 2009; Margulis et al., 2015), here we are content with presenting the analysis equation for the posterior and how to implement it for the snow reconstruction

problem. Each particle represents a different annual integration of the snow model and will have a unique forcing history associated with it as dictated by the perturbation parameters described in Section 2.An overview of the tool chain is given in Figure 1.3.1. A priori, each of these histories is assumed to be equally likely. The observed fSCA depletion for the given water





year and its assumed error structure is then used to constrain the ensemble of particles through the PBS analysis. In the PBS, the Bayesian posterior is approximated by a discrete probability mass function consisting of the posterior weights of each of the particles (model trajectories). As shown in Aalstad et al. (2018), when each particle is given an equal prior weight ($1/N_e$) and a Gaussian likelihood is used, the posterior weight for the $j$-th particle is given by

$$w_j = \frac{\exp\left(-||\mathbf{d}_j||_{\mathbf{R}}^2/2\right)}{\sum_{k=1}^{N_e} \exp\left(-||\mathbf{d}_k||_{\mathbf{R}}^2/2\right)}, \tag{1}$$

where the square norm of the innovations (residuals) for an arbitrary particle $k$ is given by

$$||\mathbf{d}_k||_{\mathbf{R}}^2 = \left(\mathbf{y} - \widehat{\mathbf{Y}}_k\right)^{\mathrm{T}} \mathbf{R}^{-1} \left(\mathbf{y} - \widehat{\mathbf{Y}}_k\right) \tag{2}$$

in which $^{\mathrm{T}}$ denotes the matrix transpose, $\mathbf{R}$ is the observation error covariance matrix, $\mathbf{y}$ is the observation vector containing the remotely sensed fSCA depletion for a given snow season, and $\widehat{\mathbf{Y}}_k$ is the predicted observation vector containing the corresponding modelled fSCA for particle $k$. The particle approximation of the Bayesian posterior represented by (1) improves

as the number of particles increases. It should be clear from the analysis step (1) that by definition the posterior weights sum to one. Furthermore, unlike the ES, the PBS only changes the relative weights of the particles and not their position within the model space. This makes the PBS particularly attractive in a clustering framework as we do not need to rerun the ensemble after the analysis.

An important component of DA is the prescribed error covariance structure of the observations. Since the MODIS fSCA

retrievals that we are assimilating are affected by various error sources that vary from day to day, such as atmospheric conditions and viewing angle, we assume that the observation errors are uncorrelated in time. Moreover, we assume a fixed observation error variance $\sigma_y^2$. Thereby, we use a simple scalar diagonal observation error covariance matrix $\mathbf{R} = \sigma_y^2\mathbf{I}$ where $\mathbf{I}$ is the identity matrix in line with similar studies (e.g. Margulis et al., 2015; Aalstad et al., 2018). This simplifies (2) which reduces to a simple square sum of innovations normalized by a constant ($\sigma_y^2$). We prescribe an observation error standard deviation of $\sigma_y = 0.13$

based on the estimate in Aalstad et al. (2018) (see Section 3.3). In order to make the model trajectories comparable to the fSCA retrievals during the analysis step, i.e. to generate the predicted observations $\widehat{\mathbf{Y}}$, an observation operator is required. We use a simple threshold on the SWE to determine the binary (snow/no-snow) snow-cover of each modelled grid cell based on values from Thirel et al. (2013) while also accounting for possible surface roughness. Due to the scale difference between the MODIS pixels ( 500 m) and the model grid cells (30 m), the modelled fSCA within a MODIS pixel is then simply the average of the

binary snow cover in all model grid cells that fall within that pixel.

## 3   Data

### 3.1   Meteorological forcing

Driving climate data are obtained from the ERA5 reanalysis from ECMWF. This is the latest reanalysis from ECMWF that updates the ERA-Interim reanalysis. The main improvements are an increase of spatial resolution to 31 km, hourly temporal





resolution, and increase in vertical model levels to 137. Accumulated values are now from the last time step and not last forecast as in ERA-Interim. This means that we can easily obtain the mean rates required to drive our numerical model by simply dividing these accumulations by the hourly time step (Fiddes and Gruber, 2014). Forcing data is detailed in Table 2. For each TopoSUB cluster, defined by the mean physiographic characteristics of a cluster, (Fiddes and Gruber, 2012) the ERA5 meteorological fields are downscaled using TopoSCALE (Fiddes and Gruber, 2014).

## 3.2 Surface properties

TopoSUB requires topographical parameters as input predictors to the clustering algorithm. We derive the following topographic parameters from the SRTM-3 digital elevation model: elevation, slope, aspect and sky view factor (proportion of visible sky). Surface cover is characterized in a simple 3 mode classification in order to approximate sub-surface stratigraphies: first a threshold on MODIS NDVI is used to classify vegetated surfaces, then a simple model further differentiates between steep bedrock and debris slopes. Further details are available in Fiddes et al. (2015).

## 3.3 Assimilated fSCA observations

We assimilate fSCA retrievals obtained from version 6 of the level 3 daily MODIS snow-cover product from the Terra (MOD10A1 product; Hall and Riggs, 2016a) and Aqua (MYD10A1 product; Hall and Riggs, 2016b) satellites. The retrieval algorithm is based on the inversion of a linear regression of MODIS normalized difference snow index (NDSI) on reference fSCA estimated from coincident Landsat imagery and it is given by the 'FRA6T' relationship in Salomonson and Appel (2006). The normalized difference snow index exploits the fact that snow is highly reflective in the visible but a good absorber in the shortwave infrared which differentiates it from most other natural surfaces (Painter et al., 2009). If cloud free retrievals are available from both Terra and Aqua retrievals for a given day then the Terra retrievals are used. Aalstad et al. (2018) compared MODIS fSCA retrievals to reference fSCA estimates obtained from a time-lapse photography, imagery from an unmanned aerial vehicle, as well as snow surveys at a site on Svalbard and obtained an RMSE of $\sigma_y = 0.13$ for the MODIS retrievals. This estimate is in reasonable agreement with those found at other sites (e.g. Mason et al., 2018), and so we use this as the as the observation error variance ($\sigma_y^2$) in the assimilation (Section 2.3.2).

## 3.4 Evaluation

### 3.4.1 Station data

SWE (mm) measurements obtained manually by observers are available at approximately biweekly intervals from snow profiles across Switzerland. Here we use the GCOS dataset which consists of 11 sites (Figure 2). We call these sites 'stations' throughout the paper. The dataset is openly available (Marty, 2017). Automatic HS (cm) measurements performed by sonic ranger (Campbell Scientific SR50) are available from the Intercantonal Measurement and Information System (IMIS) station network at 30 minute intervals. This is a high elevation station network that forms the backbone of the national avalanche service in Switzerland.




### 3.4.2 Airborne snow height retrievals

The Airborne Digital Sensor (ADS) opto-electric line scanners ADS80 and ADS100 from Leica Geosystems were used to acquire summer and winter stereo images which were processed into high resolution digital terrain models (DTM) using photogrammetry (Bühler et al., 2015). HS is then retrieved by subtracting summer from winter DTM and available for two footprints in the Davos region covering the Wannengrat area ( $3.5 \times 7.5$ km) and the Dischma area ( $7 \times 17$ km)) of high alpine terrain. The footprint of this survey is shown in Figure 2. These data are used for spatial evaluation of the scheme. Acquisition dates are 20 March 2012, 15 April 2013 and 17 April 2014. All snow depth maps were calculated using a summer DSM from 3 September 2013. The resolution of this dataset is 2m with a vertical RMSE of around +/- 30 cm (Vögeli et al., 2016; Bühler et al., 2015). The datasets are resampled to 100m (Vögeli et al., 2016) and used here to evaluate the methods. Snow depth in areas covered with forest, scrub, buildings and water bodies can not be determined using the ADS (Bühler et al., 2015) and are therefore masked out from the datasets. This dataset is openly available (Vögeli et al., 2016). Additionally as only a single summer (2013) DTM was used, all glacier areas were masked out to avoid errors associated with changing glacier surfaces. Glacier outlines were obtained from the GLIMS repository (Raup et al., 2007).

## 4 Experimental setup

In this study we conduct experiments at various spatio-temporal scales in order to comprehensively test the framework and assess its suitability for various applications. The experimental setup is shown in Figure 2. Simulations are run in 9 ERA5 grid boxes spanning the Swiss Alps. Each grid box contains at least 1 SWE measurement location and additionally several IMIS stations that are used to evaluate HS results. In addition we perform large area simulations on the entire Swiss Alps domain to explore how seasonal extremes are represented at large scale.

A prerequisite to the first two experiments (Section 4.1-4.2) is the PBS analysis step as described in Section 2 which generates the posterior weights matrix $\mathbf{W}_p$ based on PBS analysis units of MODIS cells. This then has dimensions $N_e \times N_p$ where $N_e$ is the number of MODIS pixels and $N_p$ is the number of particles (ensemble members). The following describes how $\mathbf{W}_p$ is used to generate posterior estimates of a given state variable (SWE or HS). The third experiment (Section 4.3) differs in that the analysis unit is the ERA5 grid cell itself and aims to correct aggregated grid level bias in forcing.

### 4.1 Point DA

Point scale DA is accomplished by simply mapping the DEM cell corresponding to point of interest (e.g. a validation station) to the corresponding MODIS pixel for that location. The $\mathbf{W}_p$ derived from the PBS analysis for that MODIS pixel is then used directly to generate the posterior estimate for that point. Model state results are obtained from the TopoSUB cluster that the DEM cell is a member of. Cumulative distributions are computed through the ranking of the ensemble of state variables followed by a cumulative summation of the correspondingly sorted weights. These distributions allow for the estimation of quantile values of the posterior model state.





## 4.2 Spatially distributed DA

The particle batch smoother has typically been applied at point scale or on regular grids. Here we generalise the method so as to fit the TopoSUB approach. The basic aim is to generate posterior weights for each TopoSUB cluster (which has no location, only physiographic attributes) so that the weights can be used in a highly scalable and powerful manner to generate different products such as a timeseries of the posterior median aggregated to give basin level statistics. The key challenge in this aim

is how to map the spatial unit of the PBS algorithm, the MODIS pixel, which has a location in space, to a TopoSUB cluster which does not. We achieve this through the following

$$\mathbf{w}_c = \mathbf{W}_p \cdot \mathbf{a} \tag{3}$$

where $\mathbf{W}_p$ is the $N_e \times N_p$ weights matrix and $\mathbf{a}$ is a $N_p \times 1$ vector containing the fractional abundance (cover) of cluster $c$ represented in each of the MODIS pixels. Here $\mathbf{w}_c$ contains the weight of each forcing history for cluster $c$ and is computed

for each of the clusters. This yields the weights matrix $\mathbf{w}_c$ that contains the weights of the forcing histories for each cluster. As a second step the weights $\mathbf{w}_c$ are renormalized to sum to one since that is not guaranteed in Eq. 3.

## 4.3 Coarse grid DA

The third DA method addresses bias in forcing at grid level only, it is the most efficient and lightweight of the three approaches. It also differs from the previous methods in that the PBS analysis step is computed at ERA5 grid unit not MODIS pixel unit. This

makes the analysis step highly efficient and scalable over large areas. It is emphasised that while the two previous methods address both aggregated bias in forcing at grid level they also correct errors in the subgrid method (such as physiographic description) and downscaling (such as precipitation distribution), this method only corrects the bias at grid-level. However it is of interest if we seek a simple and robust way to feedback subgrid information to large scale atmospheric grid cells, in this case using ERA5:

1. Compute MODIS fSCA aggregated to the large scale atmospheric grid cell (ERA5) while accounting for clouds (max 10 % cloudiness tolerated). Cloud pixels are filled with mean fSCA value of the cluster to which the pixel belongs.

    2. Compute the predicted observations, i.e. the modelled fSCA, for each cluster and aggregate these to the ERA5 grid cell scale by multiplying by cluster members.

    3. Run PBS at ERA5 grid level to generate a single weight vector for the ensemble.

## 4.4 Run Configurations

All runs are performed using 100 particles, 150 TopoSUB clusters and cover the period 1 September 2011 - 1 September 2017. The specific temporal period covered by a given result is defined in the text. Throughout the paper a single year refers to the year in which melt occurs, e.g., "2012" refers to the period 1 September 2011 - 1 September 2012. These "water years" are prefixed with WY (e.g. WY2012) to avoid ambiguity. We measure computational effort through the number of GEOtop



model runs ($N_r$) required per year in an ERA5 grid cell. Recall that the ERA5 grid cell is the fundamental unit on which the downscaling and clustering is performed. In terms of the number of clusters ($N_s$) and particles ($N_e$), this effort becomes

$$N_r = N_e \times N_s. \tag{4}$$

In the case of the configuration used in this study ($N_e = 100$, $N_s = 150$) this amounts to $1.5 \times 10^4$ individual model runs. At 30 m resolution, there are $10^6$ model grid cells within a single $0.25°$ ERA5 grid cell. So, an explicit fully distributed simulation

with 100 particles would require $N_r = 10^8$, a four order of magnitude increase in computational effort relative to the setup used in this study.

## 5 Results

### 5.1 Evaluating the forward model

Figure 3 shows performance of the forward model at the Weissfluhjoch (WFJ) research site (see Figure 2) assessed over the
period WY2012-2017. It illustrates the performance of the downscaling routine in providing an adequate forcing to the model (forcing bias) and performance of the model in simulating the target variables SWE and HS when driven by downscaled ERA5 reanalysis (revealing model and forcing errors) and station observations (revealing model and observation errors). It shows that the TopoSCALE downscaling routine does a reasonable job of providing forcing to the forward model (top row) with the 0.71 °C RMSE for 2 m air temperatures being particularly low. Conversely, high wind velocities tend to be positively biased,
most likely as wind fields representing the free atmosphere on pressure levels have no surface drag that would be present in surface observations. Modelled HS and SWE (bottom row) are captured fairly well capturing both the onset and melt of the snowpack. However, peak values are generally negatively biased with respect to observations and station driven model runs. WY2012 is an obvious outlier with large snowfalls not captured by ERA5 precipitation. This can be seen by cumulative precipitation totals computed with and without WY2012 totals (Figure 3). This is reflected in simulated HS and SWE totals.
The performance of the forward model can be analysed by driving with station measurements to remove most uncertainty associated with driving reanalysis data (but with residual observation errors). ERA5 driven simulations are comparable or even outperform station runs in WY2013 and WY2014.

### 5.2 Point DA

In this experiment we compare the prior and (single pixel) posterior HS and SWE for WY2016 to the measured values at the
respective stations. An example at Truebsee GCOS station (Engelberg) is shown in Figure 4. This figure demonstrates the effect the assimilation has not only on the fSCA (which is assimilated), but also on estimates of the other state variables (in this case SWE) which get closer to independent observations. Here you see clearly how the posterior estimate of SWE (blue shading) is constrained by the assimilation and the posterior median (blue line) is much closer to validation SWE observations than the prior (red line). We then scaled this up to 9 ERA5 grid boxes that span the Swiss Alps and contain 11 GCOS SWE stations.
Additionally each box contains multiple IMIS stations measuring HS, which we also looked at in the interest of obtaining more





validation data. Significant improvements in the posterior were seen in the estimation of both variables (Figure 5). We found improvement in SWE was greater than that of HS. We hypothesize that this is due to representation of snow densities in the snow model. However, the improved representation of the snowpack mass balance as shown by improved SWE estimates is the main variable of interest in our approach. Stations where DA performed worse than the prior (supplement) can be attributed to poorly characterised melt seasons, lack of MODIS retrievals, and/or the MODIS retrievals not being representative at the scale

of the observations (c.f. Section 6).

### 5.3    Spatially distributed DA

We evaluated the performance of the method in improving the spatial patterns and absolute quantities over large areas using data from an airborne digital sensor which has been used to generate high resolution surfaces of HS in WY2012, WY2013 and WY2014 (Figure 6, WY2014 only). Both WY2012 and WY2014 show marked improvement in all spatial statistical

measures including the mean value, standard deviation (indicating increased variation) and error statistics such as RMSE and bias. WY2013 shows little improvement. We would expect a better performance for SWE than for HS due to the previously mentioned issues with the modelled snow density (see Section 5.2 and Figure 5). Figure 6 shows how the 90th percentile range is constrained by the analysis going from the prior to the posterior. Figure 7 shows probability density distributions for observations, prior and posterior in WY2014. The shape and moments such as the mean more closely match the observations

in the posterior distribution. However, the method fails to capture the very highest accumulations in the distribution (> 2.5 m), possibly due to averaging effects of generalising weights to TopoSUB clusters.

### 5.3.1    Interannual validity of weights

We tested the ability of weights obtained in a given hydrological year to improve results in a different year. We did this by looking at statistics on the Dischma basin through a cross validation exercise where each year was forced with results from

the two other years (WY2012, WY2013, WY2014). Posteriors forced by weights of other years improved performance over priors in all cases (Figure 9). This suggests that the DA method here also works to correct errors that are consistent from year to year. This could be related to spatial patterns of melt, a consistent bias in the forcing or errors in the model itself. This is an interesting result that suggests that while this method is primarily a post processing method it could be used to improve now/forecasts by using previous year weights. Additionally an analogue approach could be used to find years of best fit to

current season in order to select weight sets (Kolberg and Gottschalk, 2010).

### 5.3.2    Large-scale application: Seasonal variability

December 2016 was an extremely snow poor month and start to the winter season. Many ski areas throughout the Alps could not open until late January due to lack of snow. We compare this to December 2011 which was relatively snow rich with above average precipitation and average temperatures for the month (cf. www.meteoswiss.admin.ch). Specifically, we investigated

how open-loop runs in two contrasting seasons compared to observed spatial patterns of fSCA from MODIS and SLF reports





dated 22 December of respective year (Figure 10). The model was found to compare well with both spatial patterns of fSCA and SLF snow depth maps, which are an operational product created by interpolating station data constrained by AVHHR observed snow extent. Both fSCA and HS show snow free zones deep into alpine valleys during December 2016, and absence of snow over the northern regions and Jura Mountains. The red box indicates domain of DA runs for these seasons shown in Figure 11.

5    Next we zoomed in and compared spatial patterns of HS between the deterministic open loop and Posterior run which demonstrated that DA has increased elevation gradients of variability by reducing HS in valley bottoms and increasing it on higher slopes. DA as mentioned previously, therefore has the effect of increasing variability in the snow cover distribution. Snow cover extent estimation is also improved by DA with increased snow free area in valley bottoms showing improved fit to MODIS observations.

### 10 5.4 Coarse scale DA

Aggregated series of observed and modelled fSCA are computed at ERA-grid level. This could equally be a hydrological unit such as a basin. The main idea is to correct grid level biases in the forcing only. If we assume this is the main source of uncertainty, especially with a view to correct large scale biases, this is an effective method to apply at the scale of meteorological reanalysis. Data from WFJ is used to illustrate this point. Figure 8 shows two contrasting snow season WY2012 (high) and 15 WY2014 (low) where mean snow depths and SWE differed by a factor of 2, as recorded at WFJ (Figure 3). We compare the total ERA5 precipitation (PSUM) over the winter period Jan-April 2014 (400 mm) and compare to totals recorded over the same period at WFJ station (350 mm), ERA5 captures WFJ totals well. It should be added that there is some elevation difference between the ERA5 grid (2024 m asl) and the WFJ station (2560m asl). However, in WY2012 we see quite a different story. The ERA5 grid gives us slightly higher PSUM values of 440 mm whereas the measured PSUM was almost double this at 826 20 mm. Figure 8 shows how grid level biases in the driving forcing from ERA5 have been successfully decreased in WY2012 resulting in increased SWE totals, whereas in WY2014 where ERA5 performance was much better (cf. Figure 3), DA has had a negligible effect. This simple approach is an extremely cost effective method of assimilating slope scale subgrid information (in this case fSCA) to correct coarse grid scale forcings (ERA5). It is additionally generic enough that it could be used with various other subgrid observations such as soil moisture, to improve grid level responses.

### 25 6 Discussion

In the following discussion some emphasis is placed on sources of uncertainty arising due to generally unknown errors in both the model, observations and forcing. These errors can be systematic (bias) or random as well as errors of representativeness (e.g. Lahoz and Schneider, 2014). Accounting for the uncertainty that results from these errors is an important component of any DA framework.



## 6.1 Sources of error

### 6.1.1 Forcing bias

In this work we encounter two forms of bias in the forcing, firstly (i) grid level inputs or bias in the forcing ERA5 reanalysis. This may exist due to e.g. a bias in assimilated observations (Synop stations tend to be in valley bottoms) or errors, omissions, parameterisations in the atmospheric model itself. How different is the forcing from observed grid averaged conditions? Of course this is a very difficult question to answer. Although with products such as precipitation radars such comparison of model and measured grid integrated precipitation may be possible. The second (ii) is error (random or bias) in the downscaling routines or disaggregation of the forcing at subgrid level. Do we get gradients along topographic correct? These sources of error could well be reinforcing or indeed cancelling, as they can be independent sources of error. In the approach of spatial DA we address both systematic and random error in the forcing but with an emphasis on the former. There is no 2D redistribution in terms of longitude and latitude position of a grid box. All members of a cluster are equally perturbed and clusters do not have x,y coordinates. In point DA, again both sources are addressed but with a stronger focus on (ii) as the data assimilation is done at MODIS pixel level and therefore redistributes precipitation not only with topographic parameters but also in a spatial x,y sense. In grid DA we only address (i) which could be a useful approach in differentiating and quantifying sources of bias as well as simply and robustly addressing the question of grid level bias.

### 6.1.2 Model error

We do not focus on structural errors in the forward model as this was not the subject of this study, and further the methods are designed to be quite independent of model type. However it is worth commenting that the majority of the results in this paper have focused on HS due to higher data availability. However Figure 5 shows that results are significantly better for SWE, possibly due to errors in the model densification parameterisations. This is however reassuring as HS results can be interpreted as conservative and therefore if we were able to validate more extensively against spatial distributed SWE measurements, we would likely see improved results.

### 6.1.3 Melt period definition

An important feature to mention and not often addressed by DA studies (Morzfeld et al., 2018, is a nice exception), is sensitivity of data assimilation methods to the observations chosen for assimilation. In the case of fSCA assimilations a melt period is defined as this is when the observations provide information about the snow depletion curve (e.g. Aalstad et al., 2018). We identify the end of the snowpack as the first day the fSCA values reach zero. There may be short increases in fSCA after this date but these will generally be late spring/summer snowfalls that are transient and melt rapidly. However, this date is the first available zero fSCA observation which does not necessarily equate to the exact date the snowpack melts-out as there can be a lack of observations due to cloud cover. Therefore this should be considered a source of potential bias in the system. We then found that a fixed window of 30 days prior to this date was a simple and robust way of defining the melt period. We trialled



other methods of automatically defining the end of the "complete snowcover" period but we did not find a way to do this that could work robustly over several hundred thousand MODIS pixels. Additionally, the MODIS products are quite noisy as they are generated from an empirical relationship with higher resolution Landsat data (Salomonson and Appel, 2006). This adds to the difficulty in defining a robust, general algorithm that defines the start of the melt period. As mentioned above this is a little discussed topic but due to the sensitivity of final results to the chosen method, would certainly benefit from further research

efforts.

### 6.1.4   Scale issues in assimilation

The scale difference between validation data (station or snow profile) and fSCA retrievals from MODIS creates several issues. In this study many of the sites are in valley bottoms so they are accessible on a regular basis. However, this creates a bias caused by how representative is the point measurement of the larger MODIS pixel footprint. In Alpine valley bottoms there tends to

be a lot of infrastructure, housing and rivers, which will tend to be snow free earlier (or never snow covered) as compared to the station site that will be well protected from interference allowing natural accumulation of snowfall. Therefore the MODIS footprint will tend to be observed to be "snow free" earlier than the validation point. In addition features such as a rivers, road clearance, urban heat islands are not considered in the modelling and will generate bias in the data assimilation. The most reliable sites for data assimilation, or actually we should say for validating the method, are therefore at high elevations away

from effects due to human activity or infrastructure that are not considered in the model. Figure 12 gives an example of a DA failure that is not due to the DA algorithm, this has worked well, but the representativeness of the fSCA retrievals. As you can see the posterior is pulled in the direction of what has been detected to be the main melt period (red dots). Erroneous snowfalls during late spring/ early summer are ignored as expected. End of the winter snowpack as detected by the fSCA retrievals has been correctly identified around the beginning of April. However this site is in the middle of Zermatt town and the MODIS

pixel will likely contain signal from urban effects unaccounted for by the model.

### 6.1.5   Observational errors

In addition to the scale issues, there are actual errors and cloud-induced data gaps in the MODIS retrievals. This could be incorrectly classified clouds (as snow or vice-versa) or uncertainty in the empirical fSCA algorithm. In addition the method can also suffer from a lack of observations due to persistent cloudiness at key points in the melt-period which will create uncertainties

during DA. It may be worthwhile to consider fSCA retrievals from different higher resolution satellite constellations such as Landsat (30 m resolution) and Sentinel-2 (20 m resolution). This would increase the chance of obtaining cloud free scenes as well as reduce representativeness errors even at resolutions as high as 100 m. Furthermore, the aggregation of higher resolution retrievals would lead to a reduction of random error. The effective MODIS footprint of individual pixels can be quite variable and differs markedly from the nominal 500 m pixel resolution when the view angle deviates from nadir (Dozier et al., 2008).

So, even for gridded applications, there is a considerable representativeness error in MODIS fSCA.





## 6.2 Applications

With the methods described in this paper a range of processing pipelines can be built to address a wide array of both research and operational problems. Specific strengths of the approach are:

- Slope scale forcing (climate, reanalysis, forecast) globally.

- Explicitly include the effects of high resolution topography on surface-atmosphere interactions.

- Efficient method to make large ensemble simulations feasible.

- Data assimilation to correct bias in forcing and quantify uncertainty.

Perhaps most importantly this approach allows applications to be built in remote regions where dense observation networks do not exist, such as High Mountain Asia or parts of North America. These capabilities allow for operational applications such as large area mass movement assessments related to dynamics of surface and subsurface processes. Driving the system with
NWP (forecast) data would allow nowcasting/ forecasting applications to be setup with a suitable assimilation framework such as the EnKF. While the assimilation of fSCA would be less informative in a sequential method (such as the EnKF), ensemble simulations would still provide a useful quantification of uncertainty.

Transient climate change studies using a combination of reanalysis and climate model data (e.g. CMIP5) would be a valuable research application based on this approach, for example quantifying dynamics of permafrost extent over large areas according
to a range of scenarios and models or generate a regional snowpack reanalysis product with projected future changes.

An important operational application and currently a great humanitarian need in many remote regions in Asia (e.g. Afghanistan/ Tajikistan) could be an operational avalanche forecast based on a snowpack model (e.g. SNOWPACK, CROCUS), driven by an NWP ensemble to generate a large area probabilistic forecast where few ground stations exist. This would be a relatively cost-effective system to deploy and give first order hazard assessment where none currently exists.

## 20  6.3  Further work

For the moderate (MODIS-like) resolution satellites we hope that products will emerge from Sentinel-3 and VIIRS to prolong and expand the MODIS record. For high resolution sensors there is a strong need for operational products that ideally combine available and emerging sources such as Landsat8, Sentinel-2. The French inter-agency initiative THEIA Land Data Centre is starting to produce Sentinel-2 based snow cover products. Additionally, improved cloud masks are needed as misclassified
clouds are potential and significant sources of error in the framework. Both too strict and too relaxed cloud masking is problematic, the former leads to throwing out valid and potentially important retrievals while the latter corrupts the signal that we are trying to assimilate (the actual snow cover depletion).

For additional datasets (other than fSCA) land surface temperature can be retrieved from both MODIS and Landsat and provide a means to constrain uncertainty in the surface energy balance. However, the current MODIS products are coarse at
1 km and therefore not ideal for mountain regions. Snowmelt status (i.e. binary melting/not melting) from synthetic aperture



radar (SAR) e.g. Sentinel-1 has potential to constrain uncertainty in fSCA during cloudy periods. There is also potential from ICESAT2 which could provide a way to constrain snow depth directly.

Assimilation of sparse point data could be an important extension of this work to provide means to assimilate data sources such as ICESAT2 but also be used to improve TopoSCALE by assimilating point data (stations) to improve the downscaling of reanalysis data. This could be interesting as where TopoSCALE performs most poorly is in valleys, where surface effects are poorly represented by the atmospheric model and this is precisely where stations tend to be most abundant globally. For real-time applications in remote regions extending the method to assimilate sparse observations is important as fSCA is known to have limited value in sequential (i.e real-time) data assimilation. We have shown there is some interannual validity of results in our limited test-case suggesting that systematic biases relevant to real-time applications could be addressed through reanalysis. Additionally, by creating a long term library of "best possible" reconstructions/reanalyses then training an "analogue ensemble" or even a more machine learning type approach like neural nets, could be promising.

## 7    Conclusions

In this study we have demonstrated a processing pipeline capable of producing improved land surface simulations at scale in ungauged regions. It consists of downscaling, subgrid and data assimilation components and uses only globally available datasets for both the model forcing and assimilated observations, it is therefore suitable for global applications. Specifically we have shown:

- Use of PBS data assimilation can significantly improve estimates of snowpack at various spatial scales.

- TopoSUB clustering efficiency gains make large ensemble simulations feasible.

- The methods can be used to reduce biases both at coarse atmospheric grid scale and also those related to the downscaling routines.

- The approach is suitable for regional to global applications due to efficiency and data requirements.

- A flexible set of tools allow various research and operational problems to be addressed where high resolution surface models are needed in heterogeneous terrain.

data was obtained from We propose this as a suitable method for complex model ensemble runs at scale i.e. large numbers of particles, large spatial areas or long temporal periods. Application areas include any problem where accurate slope scale forcings are required and surface atmosphere interactions need to be simulated at slope scale e.g. large area avalanche warning where the snowpack is explicitly simulated or regional-scale hazard assessment of mass movements where changing ground thermal regime is a risk factor. The toolchain can be flexibly driven by a range of forcings e.g. climate scenario data, reanalysis of past climate or real-time NWP and drive impact models for a range of domains e.g., hydrology, snowcover, soil stability or permafrost. New developments in multi-platform processing pipelines of high resolutions products from Sentinel-2 and Landsat will further improve the method in terms of representativity and availability of observations.




*Code availability.* All code used in this publication is available at https://github.com/joelfiddes/topoMAPP.

*Author contributions.* JF wrote the grant for postDoc SNF project: TopoSAT. JF and SW designed the study, KA contributed to implementation of the DA module and multiple aspects of the analysis. JF wrote the manuscript which has been contributed to and edited by all authors.

*Competing interests.* To the authors knowledge there are no competing interests.

5 *Acknowledgements.* We acknowledge the Swiss National Science Foundation project number P2ZHP2_165435 "TopoSAT: High resolution surface modelling of the Himalayan cryosphere with satellite data assimilation" and SatPerm (239918; Research Council of Norway) for providing funding for this study. We thank all data providers for making their data freely available.





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





**Table 1.** Hyperparameters (means, variances and correlations) defining the joint probability distribution from which the ensemble of multiplicative perturbation parameters are drawn. Based on Navari et al. (2016).

| Perturbed variable | Marginal | Mean | Variance | Corr($T_a$) | Corr($P$) | Corr($S^\downarrow$) | Corr($L^\downarrow$) |
|---|---|---|---|---|---|---|---|
| Air temperature ($T_a$) | Normal | 1 | 2.5e-5 | 1 | -0.1 | 0.3 | 0.6 |
| Precipitation ($P$) | Log-normal | 1 | 0.25 | -0.1 | 1 | -0.1 | 0.5 |
| Shortwave ($S^\downarrow$) | Normal | 1 | 0.04 | 0.3 | -0.1 | 1 | -0.3 |
| Longwave ($L^\downarrow$) | Normal | 1 | 0.01 | 0.6 | 0.5 | -0.3 | 1 |

**Table 2.** Description of the hourly fields obtained from the ERA5 reanalysis. All the columns headers are terms defined by ECMWF. 'levtype' refers to the level type: surf=surface, pl=pressure level. The 'type' is either: fc=forecast, an=analysis, or inv=invariant.

| name | shortName | levtype | type | units |
|---|---|---|---|---|
| 2 metre dewpoint temperature | d2m | surf | fc | K |
| Surface thermal radiation downwards | strd | surf | fc | $Jm^{-2}$ |
| Surface solar radiation downwards | ssrd | surf | fc | $Jm^{-2}$ |
| Total precipitation | tp | surf | fc | m |
| TOA incident solar radiation | tisr | surf | fc | $Jm^{-2}$ |
| 2m temperature | 2t | surf | fc | K |
| Temperature | t | pl | an | K |
| Relative humidity | r | pl | an | % |
| U component of wind | u | pl | an | $ms^{-1}$ |
| V component of wind | v | pl | an | $ms^{-1}$ |
| Geopotential | z | surf | inv | $m^2s^{-2}$ |





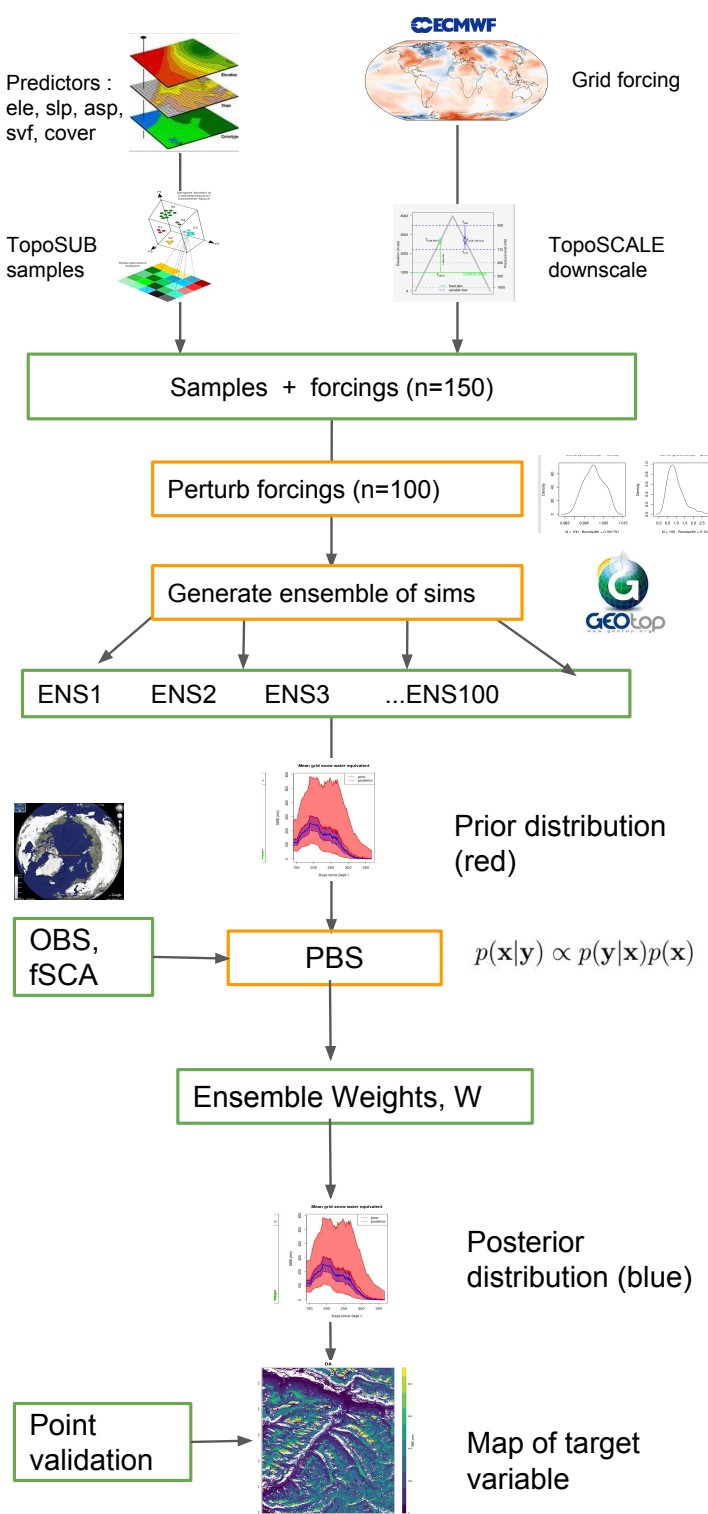

**Figure 1.** Schematic of the modelling setup.



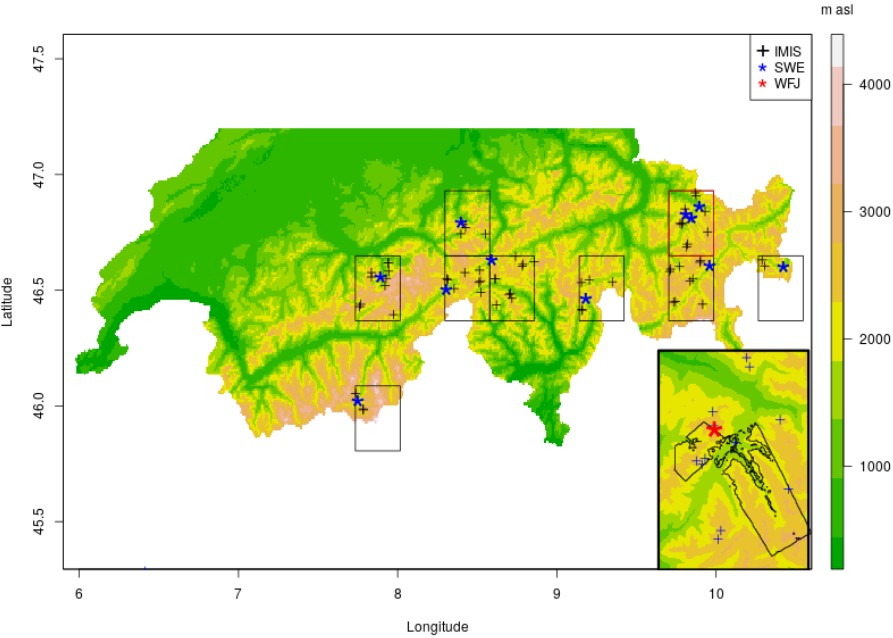

**Figure 2.** Experimental setup: 9 ERA5 grid boxes are simulated chosen for existence of GCOS SWE monitoring sites (11 stations). All IMIS stations in each box are used for evaluation (39 stations). Box in red is located the Weissfluhjoch research station as well as the flightpath of ADS data (inset).




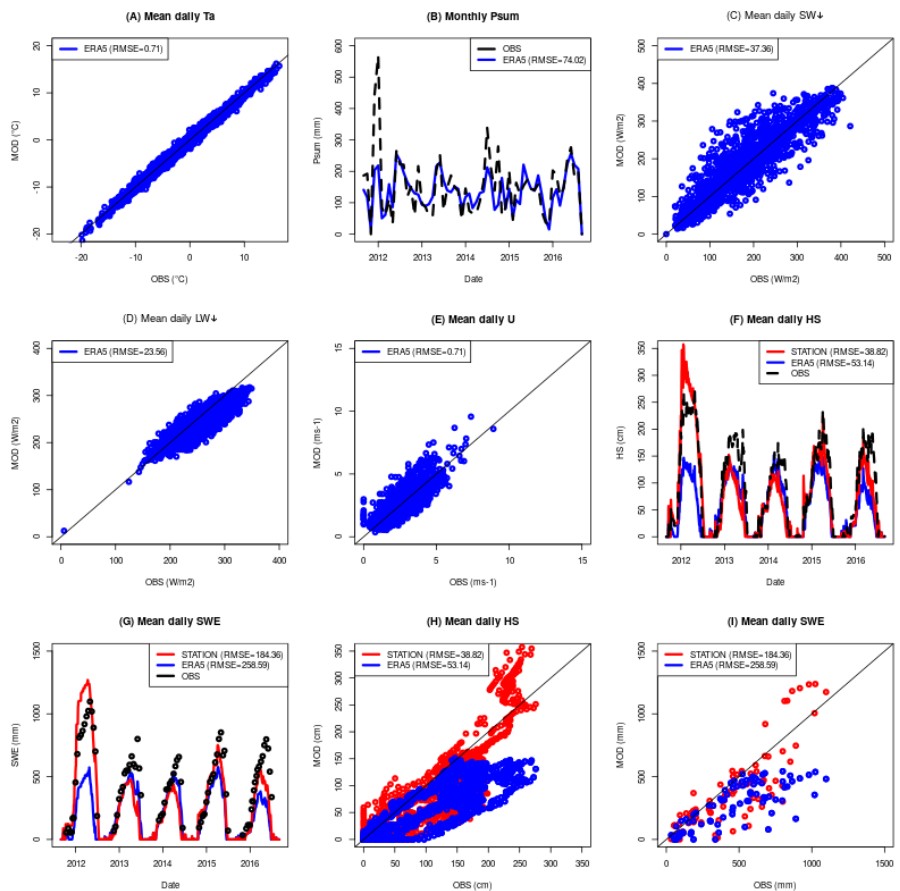

**Figure 3.** Multiyear simulations at station WFJ (WY2012-2017) in order to show baseline results for the modelling scheme. (A-E) assesses the downscaling scheme by showing downscaled ERA5 data (MOD) compared to station measurements (OBS). (F-I) assesses the simulation of target variables SWE and HS in both timeseries and scatterplots. Here, MOD is a simulation driven either by downscaled ERA5 or directly by station measurements. OBS are SWE and HS measurements made at the station. WY2012 is a clear outlier in poor performing ERA5 as shown by cumulative precipitation errors and in HS and SWE time series. HS and SWE scatter plots also show this low performance in high values attributed to WY2012. Additionally, ERA5 simulated HS is increasingly biased with depth as errors accumulate over the season to max depths. The same pattern is evident with SWE. It is worth noting that in differentiating sources of error these plots are useful. OBS - STATION approximates model error whereas STATION -ERA5 approximates the forcing error.




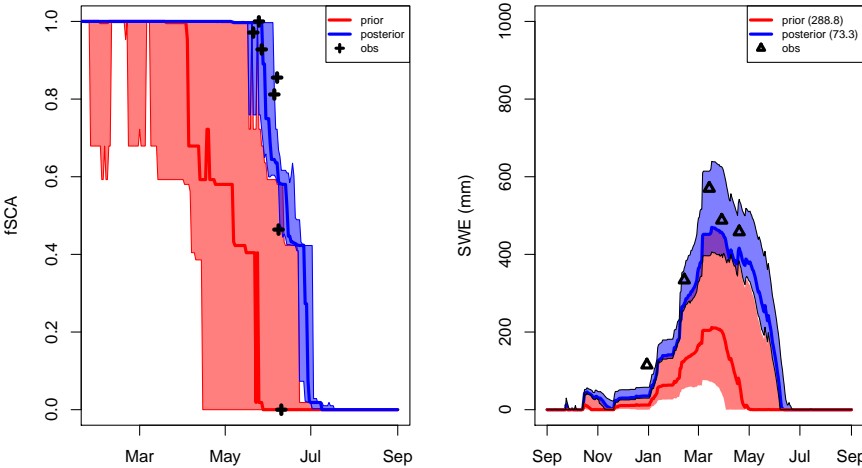

**Figure 4.** DA run at the Truebsee GCOS station, Engelberg. In the left panel is the assimilated observation, in this case fSCA represented by dots. Green dots show all observations from the combined MODIS products that are available for this stations location. Red dots indicate the observations that have been assimilated per the melt season definition. The shading and solid lines show the 90th percentile range and median of the prior (red) and posterior (blue) estimates. The right panel shows the target variable, SWE in this case. Posterior/prior are denoted in the same way. Black triangles indicate measurements used for validation.

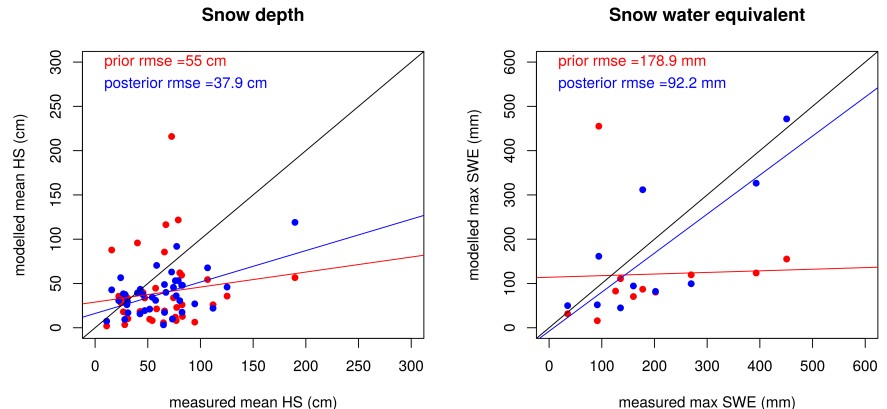

**Figure 5.** Simulated snow depth at IMIS stations (HS) and snow water equivalent at GCOS stations (SWE) for both the prior (red) and posterior median (blue) compared to observation mean. The mean is computed from all values over the entire WY2016. Posterior estimate is markedly improved in both variables. Regression lines compare the fit of posterior and prior estimates with respect to observations against the 1:1 line.





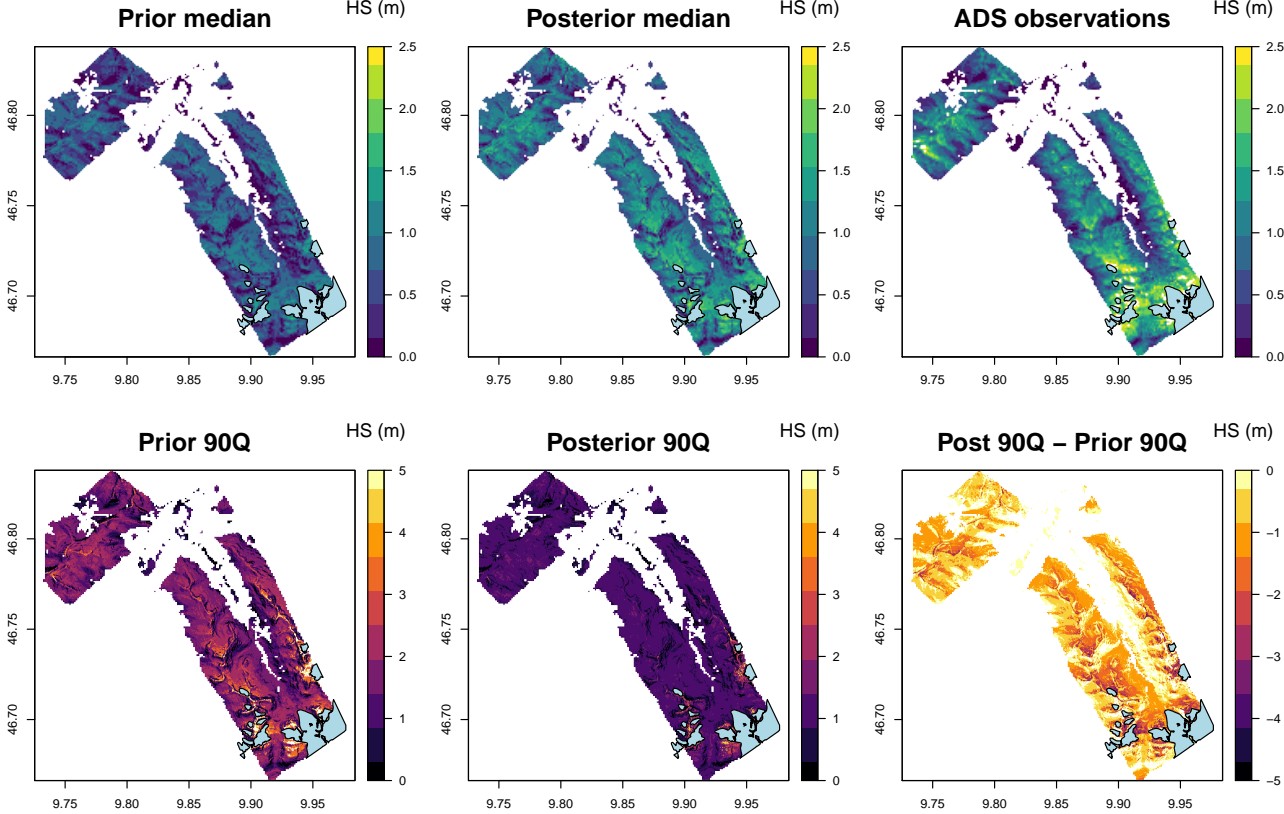

**Figure 6.** Topo row: Prior/posterior median and observations from ADS sensor flights in Davos region of 14 April 2014 (see Fig 2 for location). Bottom row: uncertainty represented by the 90th percentile range of the ensemble and reduction in uncertainty in the posterior. Glacier mask is shown in blue. Posterior median is improved with respect to observations and uncertainty is reduced by the DA scheme.





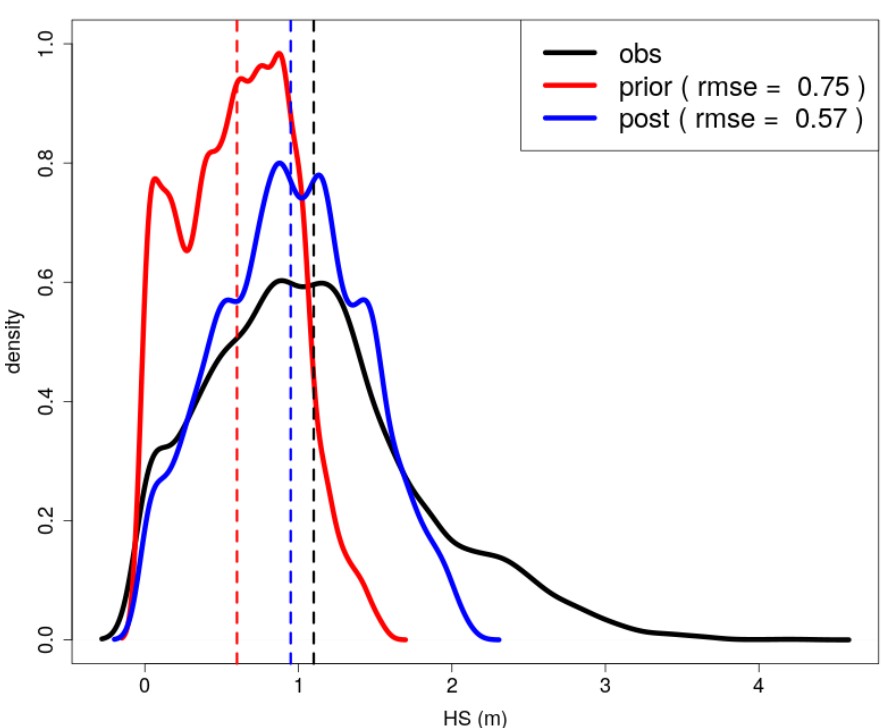

**Figure 7.** Density distribution plots for HS obs, prior and posterior within the ADS footprint for 14 April 2014 (see Figure 6). The observed distribution is better captured by the prior. Dashed lines give the respective mean values.



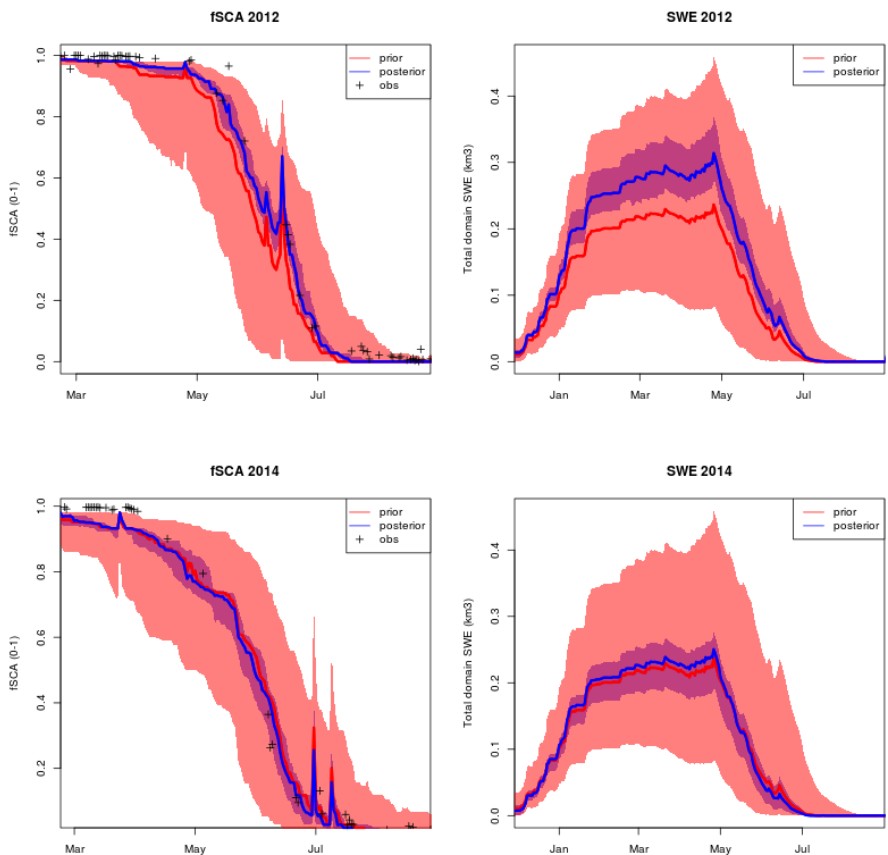

**Figure 8.** Assimilation of fSCA at grid level which targets bias in grid level forcing. Two contrasting seasons, WY2012 (top row) and WY2014 (bottom row) are shown. Vertical dashed lines give the assimilation window. Grid level biases in WY2012 are compensated for by DA. Grid level forcing was much more accurate in WY2014 and resulting effect of DA was negligible.





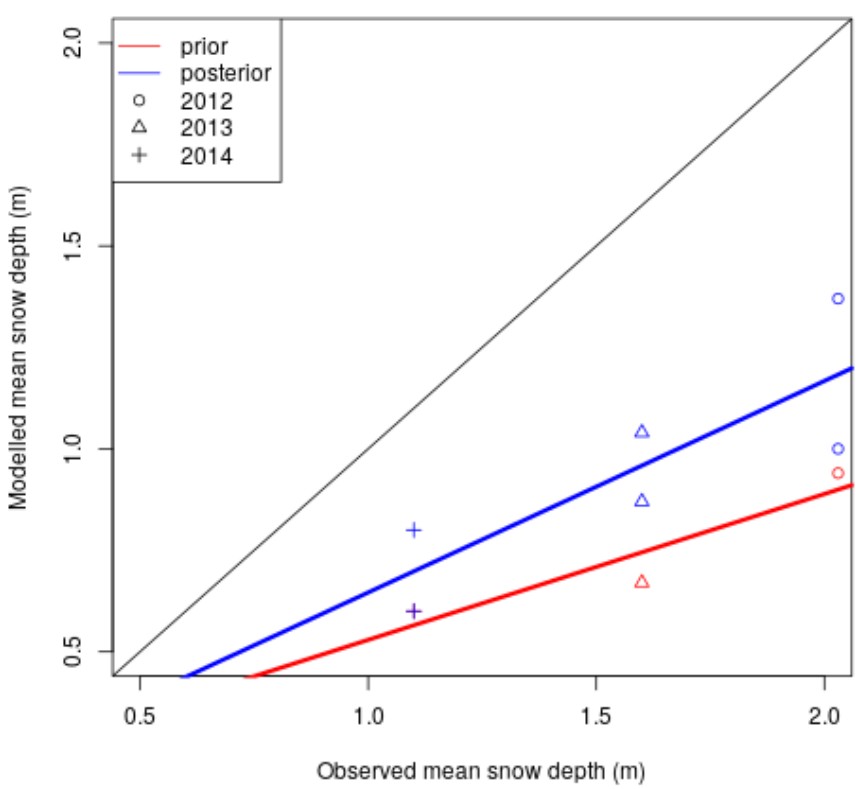

**Figure 9.** Interannual validity of weights generated by the DA scheme. Modelled versus observed mean snow depth averaged over the entire ADS zone on ADS acquisition dates WY2012, WY2013, WY2014 are shown. Posteriors (blue) are generated using weights of other two years and compared to the prior (red). For example posteriors of WY2012 are generated using weights of WY2013 and WY2014.





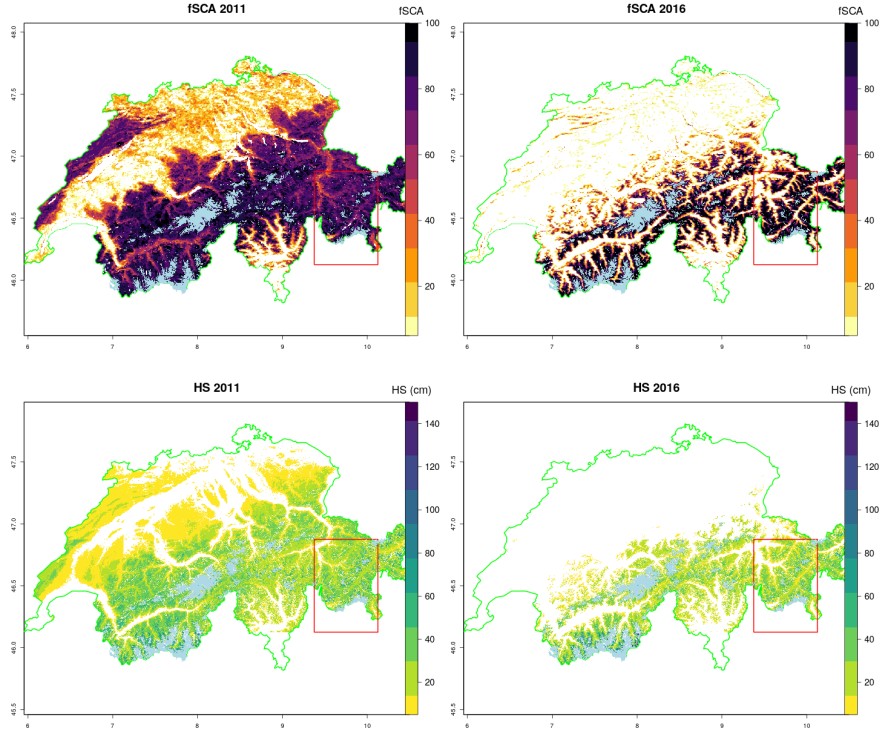

**Figure 10.** Mean December high resolution (30 m) large area HS simulations (open-loop) in two contrasting seasons (bottom) compared to observed spatial patterns of fSCA from mean December MODIS retrievals (top). Modelled HS compares well to spatial pattern of fSCA. December 2016 was an extremely dry start to the season with many ski resorts unable to open until late January. Both observed fSCA and modelled HS show snow reflect this fact with snow free zones deep into alpine valleys. Red box indicates domain of DA runs for theses seasons shown in Figure 11. Glacier mask given in light blue.




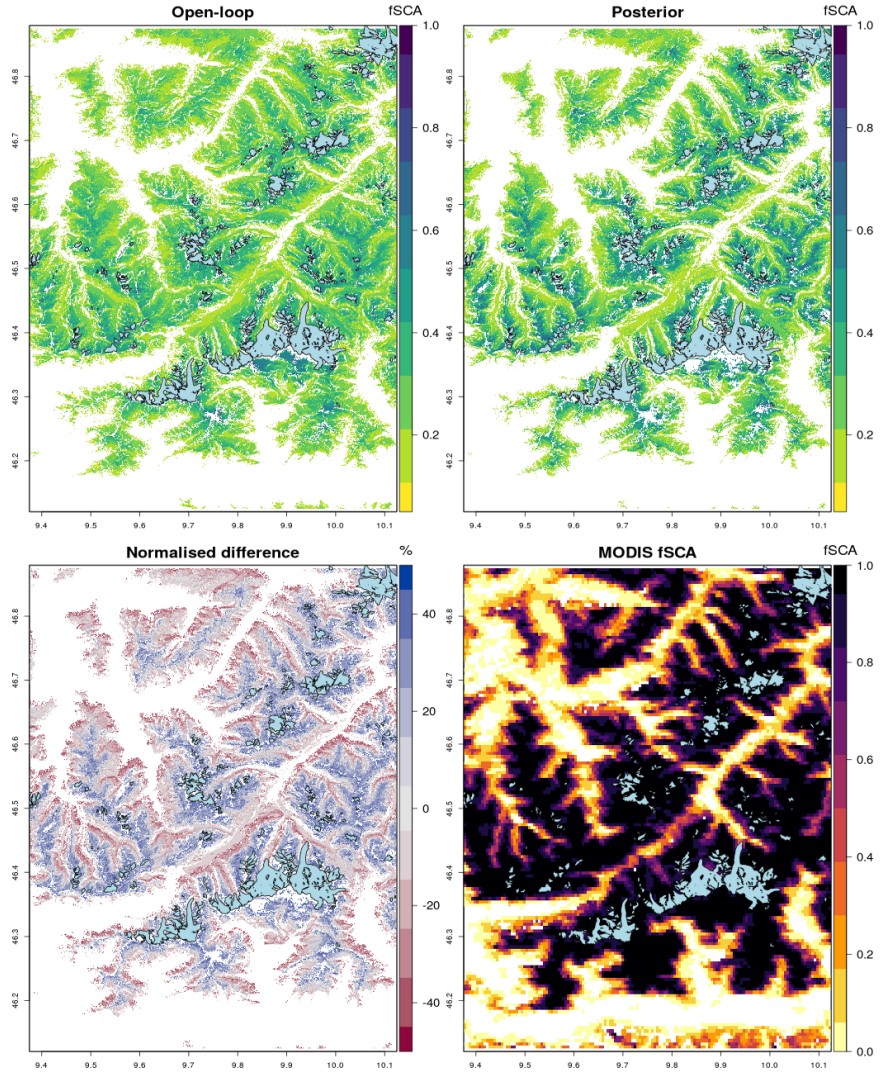

**Figure 11.** Mean HS December 2011 DA is compared to open-loop. Difference plot shows how DA has reduced low elevation snow height and increased high elevation snow height. Variability has been increased. Snow free valley bottoms show improved match to MODIS OBS. However both open–loop and posterior capture spatial patterns of snow cover reasonably well.



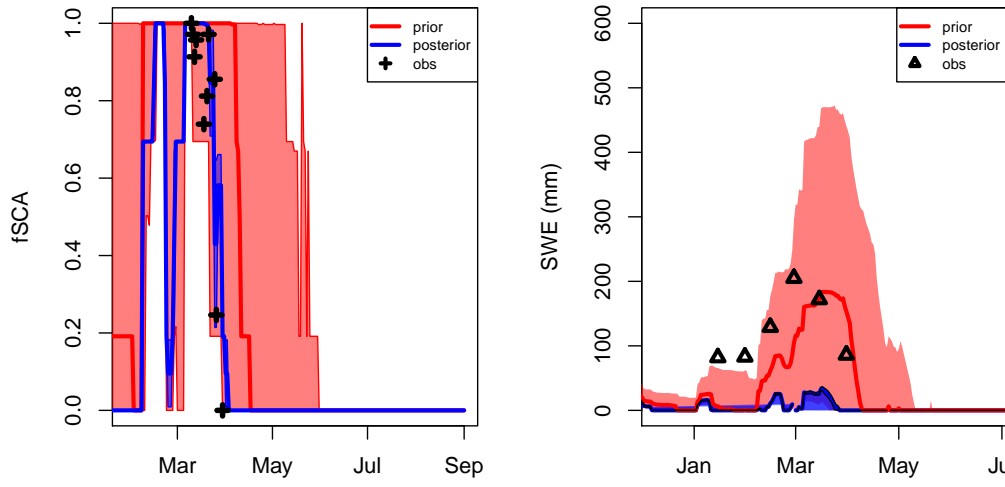

**Figure 12.** An example of poor performance due to non-representative fSCA retrievals. The posterior has been correctly pulled back to the observed depletion curve. However it is likely that the depletion curve does not well represent the validation station due to urban effects.