# Peer review of "Hyper-resolution ensemble-based snow reanalysis in mountain regions using clustering"

_Hydrology and Earth System Sciences, 2019_

## Referee Comment (RC1) · Richard L.H. Essery (Referee) · 9 Jun 2019

Fiddes, Aalstad and Westermann present interesting results on assimilation of remote snow cover observations in an efficient model of snow accumulation and melt over complex topography. It is not entirely true, as stated in the abstract, that "grid-based models cannot be run at spatial resolutions to explicitly represent important physical processes" – there are numerous examples in literature of models representing multiple physical processes being run on high resolution grids or triangular networks – but these models certainly are not optimal and cannot be run for large areas or long periods.

page 1, line 10

What are "surfacecheck models"? The abstract should say something about what data

are assimilated.

page 2, line 13

Data assimilation in land surface modelling schemes has been around for longer than might be suggested by citing a 2012 review. The North American Land Data Assimilation System was initiated in 1998, and the ECMWF model has had operational assimilation of snow depth observations since 1987.

page 5, line 32

Reference to Figure 1.3.1 should be Figure 1.

page 6, line 3

Ne is not explained. It is later described as a number of pixels on page 8 and a number of particles on page 10.

page 6, last line

ERA5 resolution was earlier stated as 25 km.

page 8, line 10

I don't think that Vögeli et al. (2016) says anything about the open availability of the airborne snow height retrievals.

page 10, line 14

A positive bias of high wind velocities is not very apparent in Figure 2.

page 11, line 15

Because fSCA contains no information about HS after it reaches 100%, the method might be expected to fail for the very highest accumulations.

page 12, line 14

This first reference to Figure 8 is out of sequence.

page 16, line 23

Delete "data was obtained from"

Table 1

Means and variance lack units

Figure 2 caption

Delete "simulated" in the first sentence. The second sentence is ungrammatical and needs to be rewritten.

Figure 3

Why is there a point with LW close to zero in (D)? STATION should be explained in the caption.

Figure 4

The green dots described by the caption are black crosses in the figure, and there are no red dots. Are there any HS observations that would help to resolve the disagreement between the posterior and the last fSCA observation?

Figure 7

Why do all of the distributions extend to negative snow depths? "The observed distribution is better captured by the posterior"

Figure 8 caption

There are no vertical dashed lines in the figure.

Labels on several of the figures are too small.
* * *
37, 2019.

---

## Referee Comment (RC2) · Simon Gascoin (Referee) · 17 Jul 2019

This paper presents a new data assimilation framework to generate snow reanalysis based on MODIS snow products. The method was tested using various datasets in Switzerland and the results are very convincing. The novelty is not the data assimilation scheme (particle batch smoother, Margulis et al. 2015) or the model (GEOtop, Endrizzi et al. 2014), but the method to optimize the numerical cost of the data assimilation by using a clever spatial reduction of the simulation domain by topographic clustering. A limitation of the clustering approach is that it does not allow an explicit representation of lateral transport (avalanches and wind transport), whereas these processes are known to be significant at "hyper-resolution" (here 30 m). In addition, if I correctly understood it implies that the DA assimilation can mix the contribution of spatially remote pixels in

the weighting of the particles.

The data assimilation pipeline was implemented at three different "experimental scales". I agree with the first referee that the approach is interesting and that previous work may be better acknowledged. I do not have any major comment (but many minor comments, see below), except that in my opinion the paper would have been easier to read if only one of the DA approaches was described and evaluated (including at different scales). In particular, the "coarse scale DA" was only briefly illustrated while it is in my opinion the most promising approach.

P1L18: "Spatial resolutions of 100 m are commonly recommended for modelling of land surface variables such as snow cover or surface temperature in complex terrain": the authors may also check Baba et al. (2019) where we specifically studied this topic (see below).

P3L1: what is hyper-efficient?

P3L6: "earths surface"

P4L25: this idea was surely introduced before 2018

P5L14: these parameters were obtained in Greenland. This should be explicitly stated in the method and discussed later.

P5L27: I do not understand why the TopoSUB approach is not compatible with an iterative approach and sequential resampling of the particles.

P6L23: Thirel et al. (2013) do not use a threshold to convert SWE to SCA but the snow depletion curve of Zaitchik & Rodell (2009). This point should be clarified.

P7L7: It is odd to derive the MODIS error from a study in Svalbard while there are multiple evaluation studies of MODIS snow fraction in temperate alpine regions which are more similar to Switzerland including the original paper by Salomonson and Appel (2006).

P7L21: Masson not Mason

P7L21: "as the as"

P8L28: cumulative distributions of what?

P10L5: It is not sure if there is a need to run a snow model at 30 m resolution especially if it does not represent wind transport and avalanches. We showed that a 250 m resolution can be sufficient to capture the main energy balance processes (Baba et al. 2019). If the resolution is set to 300 m then $N_r$ becomes $10^6$.

P12L2: "which are an"

P12: 400 mm, 350 mm and 826 mm, it may be a coincidence but why not using the same precision?

P14L2: the noise does not come from the NDSI-SCF relationship

P14L14: another important limitation is the poor accuracy of the MODIS product in dense forest areas. In particular, I wonder if it could be the cause of the DA failure observed near Zermatt rather than the "urban effect". In any case the consequence of the lack of reliable snow detection in dense forest areas must be discussed since the DA scheme is presented as applicable at global scale.

P15L23: The reference for the Theia snow products is Gascoin et al. (2018).

P15L30: "1 km not ideal" but the results show that 500 m is useful.

P16L23: "data was obtained from We"

P17: I tried to explore the code in the Github repository but it contains tens of R and Python files from multiple projects; it would be a great addition to the paper if the code was a bit more documented to allow reproducing the results of this paper or even better to allow other interested people using the DA scheme in another study area (just a suggestion!).

P18 Endrizzi et al. not a discussion paper

Figure 3: figure labels are too small.

Figreu 8: what does represent the spread? (full ensemble?)

Figure 11: top panels are HS not fSCA.

---

## Author Comment (AC1) · 14 Aug 2019

We thank Richard Essery for his time in providing constructive and thoughtful comments which have certainly improved the manuscript. Responses are detailed below with reviewer comment (RC) followed by an authors response (AR), in each case. Bold text indicates text sections that have been changed in the manuscript.

RC0: "Fiddes, Aalstad and Westermann present interesting results on assimilation of remote snow cover observations in an efficient model of snow accumulation and melt over complex topography. It is not entirely true, as stated in the abstract, that "grid-based models cannot be run at spatial resolutions to explicitly represent important physical processes" – there are numerous examples in literature of models representing multiple physical processes being run on high resolution grids or triangular networks – but these models certainly are not optimal and cannot be run for large areas or long periods."
AR0: We have qualified this statement as follows:
**"Spatial variability in high-relief landscapes is immense, and grid-based models cannot be practically run at spatio-temporal resolutions that explicitly represent important physical processes at scale."**

RC1: page 1, line 10 What are "surfacecheck models"?
AR1: typo,  "surfacecheck" -> "surface"

RC2: The abstract should say something about what data are assimilated.
AR2: We have added the following text to abstract l.8:
**"We demonstrate marked improvements in estimating snow height and snow water equivalent at various scales using this approach that assimilates retrievals from a MODIS snow-cover product."**

RC3: page 2, line 13 Data assimilation in land surface modelling schemes has been around for longer than might be suggested by citing a 2012 review. The North American Land Data Assimilation System was initiated in 1998, and the ECMWF model has had operational assimilation of snow depth observations since 1987.
AR3: Thanks for this, the sentence is currently misleading. We actually intended to refer to the high resolution surface community i.e. hydrologists or others working on surface processes such as snow deposition. LSM is obviously a term strongly connected to the climate/NWP communities but we need a suitable term for complex "surface models" such as CROCUS, SNOWPACK or GEOTOP. We have changed the text as follows:

**"While DA has a long history as a tool employed in NWP (cf. ECMWF, NLDAS), only relatively recently has DA started to be utilised in high resolution surface modelling schemes (Liu et al., 2012), but it has already shown much promise in the current era of plentiful remote sensing data."**

RC4: page 5, line 32 Reference to Figure 1.3.1 should be Figure 1.
AR4: Corrected in text.

RC5: page 6, line 3 Ne is not explained. It is later described as a number of pixels on page 8 and a number of particles on page 10.
AR5: This is a mistake, to be clear Ne = N particles , Np = N MODIS pixels and Ns= N Toposub clusters. We have added the definition on first mention (p6) and corrected occurence on p8.

RC6: page 6, last line ERA5 resolution was earlier stated as 25 km.
AR6: This is a mistake, the original grid of the model is 31km. We downloaded the netcdf product which is reprojected to a regular long/lat grid according to user specification. We set this at 0.25 degrees to match the original grid resolution. We have edited this for consistency throughout the text.

RC7: page 8, line 10 I don't think that Vögeli et al. (2016) says anything about the open availability of the airborne snow height retrievals.
AR7: This is a data citation issue,therefore not always standardised. The data used in Vögeli et al. is available here: https://www.envidat.ch/dataset/10-16904-23.
However, the authors request on that landing page that the manuscript is cited if the data is used. We switch the citation for the dataset doi here as follows:
**"This dataset is openly available (doi:10.16904/23)."**

RC8: page 10, line 14 A positive bias of high wind velocities is not very apparent in Figure 2.
AR8: This figure was generated with development code which included a wind correction algorithm - which is intended for another paper. We have regenerated the figure with the original TopoSCALE algorithm that preserves the bias. This code development was concurrent with manuscript submission and not at that stage ready to be described or properly evaluated.

RC9: page 11, line 15 Because fSCA contains no information about HS after it reaches 100%, the method might be expected to fail for the very highest accumulations.
AR9: This is true that no info is gained if there is complete complete cover (fSCA=100), however even these extreme depths ablate to fSCA=0 in our region and are therefore suitable for DA. We mask glaciers of course. We think we miss extreme values due to averaging effects as stated - therefore we prefer to leave the text as written.

RC10: page 12, line 14 This first reference to Figure 8 is out of sequence.
AR10: corrected in text

RC11: page 16, line 23 Delete "data was obtained from"
AR11: done

RC12: Table 1 Means and variance lack units
AR12: Units have been added.

RC13: Figure 2 caption Delete "simulated" in the first sentence. The second sentence is ungrammatical and needs to be rewritten.
AR13: Caption now reads as:
**"Figure 2. Experimental setup: The 9 ERA5 grid boxes were selected based on the fact that they contained GCOS SWE monitoring sites (11 stations). All IMIS stations in each box are used for evaluation (39 stations). The Weissfluhjoch research station as well as the flightpath of ADS data is located in the red outlined box, which is also shown at a larger scale in the inset."**

RC14: Figure 3 Why is there a point with LW close to zero in (D)? STATION should be explained in the caption.

AR14: This erroneous point comes from the last daily mean value in the time series which has been accidently computed from a single datapoint i.e. an incomplete day, hence the low value. This applies to all plots as can also be seen in SWin. We have cut this last daily value from all plots.

The caption is inconsistent with the latest version of the plot and has been edited as follows:

**"Multiyear simulations at station WFJ (WY2012-2017) in order to show baseline results for the modelling scheme. (A-E) assesses the downscaling scheme by showing downscaled ERA5 data (ERA5) compared to station measurements (OBS). (F-I) assesses the simulation of target variables SWE and HS in both time series and scatter plots. Here, ERA5 is a simulation driven either by downscaled ERA5 (ERA5) or directly by station measurements (STATION). OBS are SWE and HS measurements made at the station. WY2012 is a clear outlier in poor performing ERA5 as shown by cumulative precipitation errors and in HS and SWE time series. HS and SWE scatter plots also show this low performance in high values attributed to WY2012. Additionally, ERA5 simulated HS is increasingly biased with depth as errors accumulate over the season to max depths. The same pattern is evident with SWE. It is worth noting that in differentiating sources of error these plots are useful. OBS - STATION approximates model error whereas STATION -ERA5 approximates the forcing error."**

RC15: Figure 4 The green dots described by the caption are black crosses in the figure, and there are no red dots. Are there any HS observations that would help to resolve the disagreement between the posterior and the last fSCA observation?
AR15: This is from an earlier iteration in colour schemes where all (assimilated and non-assimilated) obs were shown. The submitted plot only shows assimilated obs for clarity. The text now reads:
**"DA run at the Truebsee GCOS station, Engelberg. The left panel shows the assimilation step with the assimilated fSCA observations represented by black crosses. The shading and solid lines show the 90th percentile range and median of the prior (red) and posterior (blue) estimates. The right panel shows the target variable validation, SWE in this case. Posterior/prior are denoted in the same way. Black triangles indicate the measurements used for the validation."**

Unfortunately there are no snow depth measurements at the snowpack disappearance date - the last manual observation corresponds to the SWE data in the second panel of the plot. The closest automatic station (IMIS network) is TIT2 at 2149m asl which is significantly higher than the Truebsee station at 1769m and therefore not comparable.

RC16: Figure 7 Why do all of the distributions extend to negative snow depths? "The observed distribution is better captured by the posterior"
AR16: (a) This is an artefact of the kernel density function smoothing (R function: density Figure this out. Figure this out.). We have changed the data range parameters to constrain this to 0 in the plot. (b) Sentence corrected.

RC17: Figure 8 caption There are no vertical dashed lines in the figure. Labels on several of the figures are too small.
AR17: Vertical lines have been added and labels have been enlarged.

---

## Author Comment (AC2) · 14 Aug 2019

We thank Simon Gascoin for his time in providing constructive and thoughtful comments which have certainly improved the manuscript. Responses are detailed below with reviewer comment (RC) followed by an authors response (AR), in each case. Bold text indicates text sections that have been changed in the manuscript.

RC0: The data assimilation pipeline was implemented at three different "experimental scales". I agree with the first referee that the approach is interesting and that previous work may be better acknowledged. I do not have any major comment (but many minor comments, see below), except that in my opinion the paper would have been easier to read if only one of the DA approaches was described and evaluated (including at different scales). In particular, the "coarse scale DA" was only briefly illustrated while it is in my opinion the most promising approach.

AR0: We agree that the coarse scale DA has a lot of potential which we aim to further explore in a subsequent publication. The aim of this study was to demonstrate a "proof of concept" of all three approaches which we think was important as they all have their place depending on the question being asked, e.g. if the question is site scale one would use point DA which is the most accurate but costly for large area applications. Spatially distributed DA compliments our existing efficient large area methods, whereas coarse scale DA targets only bias in the large scale forcing (arguably the most important source of error particularly if one is interested in catchment/basin scale processes such as runoff).

RC1: P1L18: "Spatial resolutions of 100 m are commonly recommended for modelling of land surface variables such as snow cover or surface temperature in complex terrain": the authors may also check Baba et al. (2019) where we specifically studied this topic (see below).

AR1: We have added this interesting reference. Nice work!

RC2:P3L1: what is hyper-efficient?

AR2: Changed to "highly efficient"

RC3: P3L6: "Earth's surface"

AR3: added the comma to "Earth's"

RC4: P4L25: this idea was surely introduced before 2018

AR4: We have cited Martinec and Rango 1981 (https://doi.org/10.1029/WR017i005p01480) one of the first studies to use snow depletion curves together with a simple snow model.

RC5: P5L14: these parameters were obtained in Greenland. This should be explicitly stated in the method and discussed later.

AR5: We have incorrectly cited these parameters, they actually originate from the study based in colorado of De Lannoy et al 2012 which in turn are based on the approach of the global study of Reichle et al. 2007. We have edited the text and caption to clarify this as follows:

**"All hyper-parameters used in generating the prior ensemble are given in Table 1 and based on values from a study in Colorado by De Lannoy et al. (2012) which in turn are based on the approach of Reichle et al. (2007), which is a global study."**

**"Table 1. Hyperparameters (means, variances and correlations) defining the joint probability distribution from which the ensemble of multiplicative perturbation**

**parameters are drawn. These parameters were obtained from De Lannoy et al. (2012) which in turn are based on the approach of Reichle et al. (2007)."**

RC6: P5L27: I do not understand why the TopoSUB approach is not compatible with an iterative approach and sequential resampling of the particles.
AR6: Poor wording on our side. TopoSUB is compatible the sentiment was that we try to build a pipeline based on efficient approaches. Changed text:
**"which would be more costly and less aligned with the efficiency objectives of the clustering (TopoSUB) framework."**

RC7: P6L23: Thirel et al. (2013) do not use a threshold to convert SWE to SCA but the snow depletion curve of Zaitchik & Rodell (2009). This point should be clarified.
AR7: We mean that we just use $SWE_{SCA=1}$ values from Thirel et al to account for surface roughness. However we use a binary approach based on this threshold. We have clarified this as:

**"We use a simple threshold on SWE to determine the binary (snow/no-snow) snow-cover of each modelled grid cell based on SWE values that correspond to full pixel coverage (fSCA=1) given in Thirel et al. (2013), this allows us to consider surface roughness."**

RC8: P7L7: It is odd to derive the MODIS error from a study in Svalbard while there are multiple evaluation studies of MODIS snow fraction in temperate alpine regions which are more similar to Switzerland including the original paper by Salomonson and Appel (2006).
AR8: The svalbard study uses an automatic camera which provides a high resolution error estimate, which arguably could be more important than the climate zone. Salomonson and Appel (2006) evaluate against Landsat pixels. The value we use is also in good agreement with RMSE's found for the standard NSIDC product by Masson et al. (0.154-0.157). We have therefore strengthened this statement as follows:
**"This estimate is in good agreement with those found in the Alps by other studies (e.g. Mason et al., 2018), and so we use this as the as the observation error variance ($\sigma 2 y$ ) in the assimilation (Section 2.3.2)."**

RC9: P7L21: Masson not Mason
AR9: Corrected

RC10: P7L21: "as the as"
AR10: removed an "as the"

RC11: P8L28: cumulative distributions of what?
AR11: changed to:
**"Cumulative distributions of state variables"**

RC12: P10L5: It is not sure if there is a need to run a snow model at 30 m resolution especially if it does not represent wind transport and avalanches. We showed that a 250 m resolution can be sufficient to capture the main energy balance processes (Baba et al. 2019). If the resolution is set to 300 m then N_r becomes 10^6.
AR12: This is a nice paper! We have since been looking at optimising the target resolution as this also reduces the TopoSUB memory requirements. We agree that wind transport and

avalanches are important processes at such high resolutions and currently investigating options to parameterise these in 1D. Note: snow is lost from steep slopes with a mass loss algorithm, however it is not redistributed as it is not clear which the downslope pixels would be.

RC13: P12L2: "which are an"
AR13: Text changed to:
**"which are operational products…"**

RC14: P12: 400 mm, 350 mm and 826 mm, it may be a coincidence but why not using the same precision?
AR14: We made a rounding error here values should be 401 mm, 350 mm, 826 mm
.
RC15: P14L2: the noise does not come from the NDSI-SCF relationship
AR15: We have edited this sentence to:

**"Additionally, the MODIS products are prone to various sources of error, as discussed below in Section 6.1.5 and this adds to the difficulty in defining a robust, general algorithm that defines the start of the melt period."**

RC16: P14L14: another important limitation is the poor accuracy of the MODIS product in dense forest areas. In particular, I wonder if it could be the cause of the DA failure observed near Zermatt rather than the "urban effect". In any case the consequence of the lack of reliable snow detection in dense forest areas must be discussed since the DA scheme is presented as applicable at a global scale.
AR16: This is a really good point, MODIS pixels even partially containing forest would definitely have an additional source of error. We have added to the discussion issues related to forest cover as a final sentence to 6.1.5 Observational errors:

**"A final important limitation of the scheme is the lack of reliable fSCA retrievals in forested areas, which applies to any optical sensor (e.g. as mentioned in the description of the ADS data)."**

However, we think our original hypothesis is still most likely as the Zermatt observer station is right in the middle of Zermatt town (see green dot on figure below) and the MODIS pixel footprint (red grid) is not contaminated by forest in this case. We have also edited the text to make it clear that this is point-scale DA which is important as the fSCA signal then comes from a single pixel and there is no risk of "forest contamination":

**"Figure 12 gives an example of point-scale DA.."**

[Figure]

Author edit: P14L30: Language correction: changed "So" to "Therefore".
Author edit: P14L16: Grammar correction: "As you can see" to "It can be seen".

RC17: P15L23: The reference for the Theia snow products is Gascoin et al. (2018).
AR17: Now corrected.

RC18: P15L30: "1 km not ideal" but the results show that 500 m is useful.
AR18: This refers to the MODIS LST product here, we do not show results for that. Surface temperatures are expected to be very heterogeneous in a 1 km footprint in mountain regions therefore it is not clear how useful assimilation would be. We have reformulated this sentence as:

**"For additional datasets (other than fSCA) land surface temperature (LST) can be retrieved from both MODIS and Landsat and provide a means to constrain uncertainty in the surface energy balance. However, the current MODIS LST products are coarse at 1 km with respect to the expected heterogeneity of LST in mountain regions (Gubler et al. 2011)."**

RC19: P16L23: "data was obtained from We"
AR19: Removed erroneous text "data was obtained from".

RC20: P17: I tried to explore the code in the Github repository but it contains tens of R and Python files from multiple projects; it would be a great addition to the paper if the code was a bit more documented to allow reproducing the results of this paper or even better to allow other interested people using the DA scheme in another study area (just a suggestion!).

AR20: We are working on python packages for this project - but as scientists first and software developers second this takes additional time!

RC21: P18 Endrizzi et al. not a discussion paper
AR21: Corrected

RC22: Figure 3: figure labels are too small.
AR22: We have increased label size.

RC23: Figure 8: what does represent the spread? (full ensemble?)
AR23: we have added the following text to the caption:
**"The shading and solid lines show the 90th percentile range and median of the prior (red) and posterior (blue) estimates."**

RC24: Figure 11: top panels are HS not fSCA
AR24: We have corrected labels and axis scaling.

**References**

De Lannoy, Gabriëlle J. M., Rolf H. Reichle, Kristi R. Arsenault, Paul R. Houser, Sujay Kumar, Niko E. C. Verhoest, and Valentijn R. N. Pauwels. 2012. "Multiscale Assimilation of Advanced Microwave Scanning Radiometer–EOS Snow Water Equivalent and Moderate Resolution Imaging Spectroradiometer Snow Cover Fraction Observations in Northern Colorado." *Water Resources Research* 48 (1): W01522.de lannoy

Gubler, S., J. Fiddes, S. Gruber, and M. Keller. 2011. "Scale-Dependent Measurement and Analysis of Ground Surface Temperature Variability in Alpine Terrain." *The Cryosphere* 5: 431–43.

Reichle, Rolf H., Randal D. Koster, Ping Liu, Sarith P. P. Mahanama, Eni G. Njoku, and Manfred Owe. 2007. "Comparison and Assimilation of Global Soil Moisture Retrievals from the Advanced Microwave Scanning Radiometer for the Earth Observing System (AMSR-E) and the Scanning Multichannel Microwave Radiometer (SMMR)." *Journal of Geophysical Research* 112 (D9): 1697.